# Merkel Cell Carcinoma—Update on Diagnosis, Management and Future Perspectives

**DOI:** 10.3390/cancers15010103

**Published:** 2022-12-23

**Authors:** Eleni Zaggana, Maria Polina Konstantinou, Gregor Herrmann Krasagakis, Eelco de Bree, Konstantinos Kalpakis, Dimitrios Mavroudis, Konstantinos Krasagakis

**Affiliations:** 1Department of Dermatology, University General Hospital of Heraklion, 71500 Crete, Greece; 2Medical School, University of Crete, 71500 Crete, Greece; 3Department of Surgical Oncology, University General Hospital of Heraklion, 71500 Crete, Greece; 4Department of Medical Oncology, University General Hospital of Heraklion, 71500 Crete, Greece

**Keywords:** Merkel cell carcinoma, immunotherapy, anti-PD-1 inhibitors, anti-PD-L1 inhibitors

## Abstract

**Simple Summary:**

Merkel cell carcinoma (MCC) is a highly aggressive skin cancer and the second cause of skin cancer death after melanoma. MCC is an immunogenic tumor. In recent years, the use of immunotherapy has changed the treatment landscape for patients with metastatic MCC, significantly improving the prognosis. However, the five-year disease-specific survival remains around 64%, underlying the unmet need for novel treatments. This review recapitulates current knowledge about MCC pathogenesis, diagnosis, and management. Emphasis is given to the use of immunotherapy and targeted therapies as well as to future therapeutic perspectives in the neoadjuvant setting and for locally advanced and metastatic MCC.

**Abstract:**

MCC is a rare but highly aggressive skin cancer. The identification of the driving role of Merkel cell polyomavirus (MCPyV) and ultraviolet-induced DNA damage in the oncogenesis of MCC allowed a better understanding of its biological behavior. The presence of MCPyV-specific T cells and lymphocytes exhibiting an ‘exhausted’ phenotype in the tumor microenvironment along with the high prevalence of immunosuppression among affected patients are strong indicators of the immunogenic properties of MCC. The use of immunotherapy has revolutionized the management of patients with advanced MCC with anti-PD-1/PD L1 blockade, providing objective responses in as much as 50–70% of cases when used in first-line treatment. However, acquired resistance or contraindication to immune checkpoint inhibitors can be an issue for a non-negligible number of patients and novel therapeutic strategies are warranted. This review will focus on current management guidelines for MCC and future therapeutic perspectives for advanced disease with an emphasis on molecular pathways, targeted therapies, and immune-based strategies. These new therapies alone or in combination with anti-PD-1/PD-L1 inhibitors could enhance immune responses against tumor cells and overcome acquired resistance to immunotherapy.

## 1. Introduction

Merkel cell carcinoma (MCC) is a rare, aggressive, neuroendocrine malignancy of the skin, first described by Toker in 1972 [1]. MCC presents a Merkel cell-like phenotype, suggesting for many years its origin from the Merkel cells of the skin. Currently, this origin of MCC is being disputed and the phenotypic similarities of MCC with Merkel cells are rather considered to result from a strong differentiation process and do not necessarily reflect cell ancestry [2,3,4]. MCC usually affects the elderly, with a median age of diagnosis between 75 and 79 years, but cases of younger, mainly immunosuppressed patients, have been reported [5]. MCC occurs predominantly in men with a male-to-female ratio of 2:1 [5,6]. The white population has a 25-times greater risk of MCC, particularly of the head and neck [6,7,8]. Immunosuppression such as transplantation, HIV infection, and hematologic malignancies is present in 10% of cases and is associated with a worse prognosis [9,10]. There is a 10-fold increase in incidence in solid organ transplant recipients and a 13.4-fold increase among HIV patients [11,12].

According to the SEER-18 Database of the National Cancer Institute in the United States, including 6600 MCC cases from 2000 to 2013, there is a 0.7 cases/100,000 person-year incidence and a 95% increase in reported MCC cases in the last years [5]. This increase can be explained by the greater diagnostic awareness of physicians, the improved immunohistochemistry diagnostic techniques (CK20 antibody use), and the aging and subsequent immune-senescence of a sun-exposed population [13]. European registries show comparable incidence rates [14], whereas the highest rates of MCC are reported in Australia [13]. MCC is the second deadliest skin malignancy after melanoma, with a case-by-case fatality rate worse than stage-matched melanomas [15]. Although its incidence is low, MCC has a strong propensity to recur locally, spread regionally to the lymph node (LN) basin, and disseminate. In most cases, patients are diagnosed with a locoregionally advanced disease [16] with an estimated five-year disease-specific survival of around 64% [17].

## 2. Disease Pathogenesis

Ultraviolet radiation (UV) and Merkel cell polyomavirus (MCPyV) infection seem to be the main oncogenic factors of MCC. The breakthrough discovery in 2008 by Feng et al. by digital transcriptome subtraction of MCPyV clonally integrated into the tumor cell genome led to a greater understanding of the pathways of oncogenesis and the molecular biology of MCC [18]. MCPyV is a commensal, ubiquitous virus acquired most frequently in early childhood, responsible for an asymptomatic life-long infection as a resident of the skin flora. MCPyV infection prevalence among adults aged 60–69 years is around 81%, with a trend of higher seroprevalence with increasing age [19]. MCPyV is considered responsible for 80% of MCC tumors [20]. The oncogenicity of MCPyV in only a small subset of infected individuals denotes the importance of additional viral truncating mutations/deletions along with a loss of immune surveillance for tumorigenesis. The virus integrates into the host DNA, driving the durable expression of two viral T-antigens, large T (LT) and small T (ST), that alter the regulation of the cell cycle, apoptosis, and other cellular pathways involved in cell transformation [21,22]. MCPyV LT specifically inhibits retinoblastoma (RB) function but cannot bind p53 [22,23]. Co-expression of ST prevents LT degradation, increases steady-state LT levels, and has been shown to enhance viral replication [24].

On the other hand, MCPyV-negative MCC have a high mutational burden with prominent UV-signature DNA damage affecting various oncogenes. Among these, mutations of the tumor suppressor genes *RB1* and *TP53,* mutations promoting the activation of the *PI3K* pathway (*HRAS*, *KRAS*), and inactivation of the Notch pathway appear to be critical oncogenic events [25]. Virus-negative MCCs have a mutational burden that is, on average, 100-fold greater than MCPyV-positive MCC [25]. It is uncertain whether a hit-and-run phenomenon with initial MCPyV integration and secondary loss of addiction to the viral T-antigens could trigger MCPyV-negative tumors [26]. Despite conflicting data, MCPyV-negative tumors seem to be more aggressive, with more frequent locoregional or distant metastasis at diagnosis and a more reserved disease-specific survival [20].

The cell of origin of MCC remains unknown. The observation of collision tumors, i.e., MCC with squamous cell carcinoma, the frequent occurrence of MCC on the head and neck region, and the high burden of somatic mutations with a UV signature suggest that MCPyV-negative cases could derive from a progenitor cell of the epidermis. In contrast, the lack of connection between tumor cells and the epidermis and the absence of a UV signature could favor a non-epithelial origin for MCPyV-positive MCC, with the most probable candidate being dermal fibroblasts [3]. A cell culture model for MCPyV infection could be established using dermal fibroblasts but not CK20-positive Merkel cells. Furthermore, only dermal cells expressing fibroblast markers were capable of expressing the MCPyV LT and ST antigens [4]. If this theory is confirmed, MCC could be the only tumor originating from two distinct germ layers: MCPyV-negative MCCs from ectodermal keratinocytes and MCPyV-positive MCCs from mesodermal fibroblasts [3]. Other putative cells of origin include pre/Pro B cells [2,3,4].

MCC is an immune-sensitive tumor, as highlighted by the association between survival and intra-tumoral levels of cytotoxic CD8+ lymphocytes [27] and of MCPyV-specific T cells [28,29], as well as the identification of specific circulating antibodies that fluctuate along with disease activity [30]. Lymphocytes of the tumor microenvironment exhibit an “exhausted” phenotype with increased expression of programmed cell death protein 1 (PD-1) and cytotoxic T-lymphocyte-associated protein 4 (CTLA-4) [31] and an increase in CD4+ and CD8+ regulatory T cells (Treg), confirming that loss of immune-surveillance is a key point in the immunoediting process of MCC [31].

## 3. Diagnosis

### 3.1. Clinical Examination

MCC usually presents as a firm, painless, nodular, or plaque-like, flesh-colored to red-violet lesion, with a high affinity for the head and neck or upper extremities (Figure 1). Rapidly growing lesions can be associated with ulceration and necrosis. Its most important features are represented by the acronym AEIOU: Asymptomatic/lack of tenderness, Expanding rapidly, Immune deficiency, age Older than 50 years, and UV exposure [9]. In a prospective study of 195 patients, the primary tumor size ranged from <1 cm in 21.3%, 1–2 cm in 43.3%, and >2 cm in 35.3% of cases [9]. The presence of nonspecific clinical features in MCC may lead to a diagnostic delay.

### 3.2. Histopathology of the Primary Tumor

A biopsy of a suspicious lesion is required for the diagnosis of MCC, followed by an immunopanel analysis. In hematoxylin-eosin staining, MCCs are dermal tumors composed of small, round, basophilic cells arranged in sheets or trabecular arrays (Figure 2). MCC cells have round, hyperchromatic nuclei with dusty (salt-and-pepper) chromatin, inconspicuous nucleoli, and scant cytoplasm with neuroendocrine granules (Figure 3). A high mitotic rate is frequent as well as vascular invasion and an abundant inflammatory infiltrate of lymphocytes and plasma cells surrounding the tumor. A band between the tumor and the epidermis is usually present. Cases of in situ MCC have been reported in the form of collision tumors [32]. Three histological patterns have been described (trabecular, intermediate, and small-cell) with no clear histo-prognostic association [33].

Diagnosis using light microscopy alone can lead to a wrong diagnosis in 60% of primary tumors and 40% of LN examinations [34]. Immunohistochemical staining is required to exclude possible mimickers, such as small-cell lung carcinoma metastasis, lymphoma, and small-cell melanoma. Tumor cells in MCC typically express markers of neuroendocrine and epithelial differentiation, such as neuron-specific enolase (NSE), CD56, chromogranin A, synaptophysin, and low molecular weight cytokeratins (CK8, CK18, CK19, CK20), CAM 5.2, and AE1/AE3 [35,36,37]. CK20 is a diagnostic marker with high sensitivity and specificity. CK20 is positive in 80% of biopsies with characteristic membranous, punctate, and/or ‘paranuclear dot-like patterns’ [35]. Another characteristic is also the absence of expression of thyroid transcription factor-1 (TTF-1) and CK7, usually present in small-cell lung cancer, of leucocyte common antigen (LCA) which is positive in lymphoma, and of S100 and HMB45, which are positive in melanoma [36]. Expression of CK20 with the concomitant absence of TTF-1 expression is diagnostic for MCC in 90% of cases [37]. CM2B4, a monoclonal antibody generated against a predicted antigenic epitope on the MCPyV-T-antigen, can be a useful reagent for the diagnosis of MCPyV-positive MCC [38].

## 4. Disease Workup and Staging

### 4.1. Baseline Imaging

After the diagnosis is confirmed by biopsy, a total body examination including LN evaluation for clinically detectable metastasis must be performed. In a recent study, the benefit of baseline imaging was confirmed, allowing the upstaging in 13.2% of cases (8.9% in locoregional and 4.3% in distant metastatic disease) among 492 MCC patients with clinically uninvolved LN [38]. In another study including 92 patients with MCC and clinically involved LN, systematic imaging revealed distant metastasis in 10.8% of cases, highlighting the further utility of baseline imaging even in the setting of clinically localized disease [39]. Thus, the National Comprehensive Cancer Network (NCCN) guidelines recommend a systematic baseline cross-sectional imaging at diagnosis [40]. The preferred imaging modalities are ultrasound of the locoregional LN, whole-body positron emission tomography (PET)-computed tomography (CT), or thoracic, abdomen, and pelvis CT with contrast, with an additional neck CT (if the primary tumor is located on the head and neck area) [40,41,42]. Brain magnetic resonance imaging (MRI) with contrast is indicated if there is clinical suspicion of brain metastasis and systematically in stage T2–T4 tumors and unknown primaries [43]. PET-CT seems to perform better than CT alone in the detection of occult disease, particularly for bone/bone marrow lesions [39,44].

Serological assessment of MCPyV-oncoprotein antibodies (against the viral capsid protein VP1) can be considered as part of the baseline workup [40]. Seropositivity can be present in almost half of MCC patients [30]. Seronegative patients are at almost 40% higher risk of recurrence and could benefit from more intensive surveillance [30]. MCPyV antibody status should be tested within the first three months of diagnosis as titers are expected to decrease after clinically evident disease is eliminated [30].

### 4.2. Evaluation of the Lymph Node Status

Occult LN metastasis can be detected in 20–40% of MCC patients, at the time of diagnosis and independently of the primary tumor size [16,16,45,46]. Thus, LN status should be determined systematically at diagnosis as it is an important staging tool and an independent prognostic factor for overall survival (OS) [17,40,41,46,47]. If LN metastasis is clinically detected, imaging studies and subsequent exploratory lymphadenectomy should be performed. All other cases should undergo a sentinel lymph node biopsy (SLNB). Ideally, SLNB should be performed alongside primary tumor-wide local excision (WLE) or precede it to avoid WLE-related lymphatic drainage modifications. SNLB evaluation should include an appropriate immunopanel analysis. SLNB-negative patients have an excellent 5-year survival of 97% [17]. False-negative SLNB results are expected in 17% of cases [48]. Even lower success rates of SNLB (45%) are typically reported in head and neck primaries, probably due to the complexity of drainage patterns [46]. If SLNB cannot be performed or is unsuccessful, the European Dermatology Forum (EDF) guidelines suggest close follow-up with clinical examination and LN ultrasound every four months, whereas NCCN guidelines propose adjuvant radiotherapy (aRT) of the draining nodal basin identified by lymphoscintigraphy [40,42].

### 4.3. Eighth Edition American Joint Committee on Cancer (AJCC) Staging for MCC and Prognostic Factors

The stage of the disease at the time of diagnosis is the most critical prognostic factor of survival [16,17,47]. The 8th edition AJCC staging system for MCC was published in 2017, based on an analysis of over 9000 patients included in the National Cancer Database, from 1998 to 2012 [47]. In this population, 5-year OS estimates for local disease (*n* = 6138), regional metastatic disease (*n* = 2465), and distant metastatic disease (*n* = 784) were 50.6%, 35.4%, and 13.5%, respectively [47]. In this latest Edition, tumor size remained an important predictor of survival, but two additional independent prognostic factors were considered. For primary tumors ≤ 2 cm, 5-year OS was improved when the negativity of LN status was confirmed pathologically versus clinically (62.8% versus 45%) [47], reflecting the obvious fact that clinically negative LN includes both node-negative and occult node-positive patients. Thus, in local disease stages I and II, clinical and pathological LN statuses were separated. Another change from the past Edition results from the observation that 5-year OS estimates for patients with “unknown primaries” and clinically and pathologically metastatic LNs were 42.2%, compared with 26.8% for those with a known primary tumor [47]. This subgroup of patients represents 5% of all MCC patients but almost 40% of patients with clinically detected LN disease. To reflect this improved prognosis, these nodal status “unknown primaries” patients were reclassified at stage IIIA (previous stage IIIB of the 7th Edition). This survival benefit is probably an indicator of an effective cell-mediated immune response responsible for the regression of the primary tumor in this subgroup [47,49].

Other parameters can have prognostic significance as well, though they are not included in the AJCC staging system [50]. With regards to the primary tumor, location on the head and neck, an infiltrative tumor growth pattern, increased tumor thickness, presence of more than 10 mitoses/high-power field, and lymphovascular invasion are associated with poorer prognosis. On the other hand, the presence of brisk tumor-infiltrating lymphocytes (TIL) is associated with improved survival. Poor prognostic factors related to the LN basin include the extent of metastatic LN involvement, large or multifocal metastatic deposits, and extracapsular extension [17]. According to a single-institution study in patients with a single positive LN, the 5-year survival rate was 66%, with 2–4 positive LNs was 62%, and with >4 positive LNs was 30% [17]. Patient-related factors associated with the worst prognosis include male sex, younger age, immunosuppression, and MCPyV seronegativity [10,30,51].

## 5. Treatment of the Primary Tumor

Considering the rarity of MCC and its complex management, patients should be referred, when possible, to a multidisciplinary team of trained specialists in reference centers, to establish a personalized treatment plan taking into consideration disease staging, patients’ general condition, and comorbidities [43].

### 5.1. Surgical Management of the Primary Tumor and the Draining Lymphatic Basin

Surgical excision followed by aRT is the treatment of choice for primary tumors [40,42]. Wide local excision (WLE) with 1–2 cm margins is proposed, mirroring melanoma practices, to guarantee disease-free margins and the removal of potential microscopic satellite metastases [40,42]. Microscopically positive margins after surgical excision relate to a higher local recurrence rate compared to microscopically negative margins (18% versus 8%) [17]. Primary closure should be prioritized, to allow the prompt initiation of aRT on the tumor bed. Complex reconstruction techniques should only be applied if necessary and after negative margins and SLNB results are obtained.

The choice of 1–2 cm-wide margins in WLE has not been evaluated in a prospective, randomized setting, but seems to be an independent factor of improved OS [52]. In a recent, single-institution study of 188 MCC cases without clinical LN involvement, surgical margins ≥ 1 cm were associated with a lower local recurrence rate. However, for patients who received aRT on the tumor bed, local control was excellent even in the case of microscopically positive margins on WLE, suggesting that, in selected cases, when tissue-sparing is mandated, narrower margins combined with aRT could be accepted [53].

A recent systematic review confirmed that Mohs micrographic surgery (MMS) is non-inferior to WLE in terms of the locoregional recurrence rate with the advantage of tissue-sparing [54] and can be applied when 1–2 cm margins are not feasible [40]. Nevertheless, in a study of 1795 patients with stage I and II MCC, MMS failed to prove an OS benefit over WLE, for now limiting its generalization as the treatment of choice for MCC primary tumors [55].

With regards to the draining LN basin, current guidelines propose a complete LN dissection (CLND) if SLNB or if the exploratory lymphadenectomy of a clinically detectable LN are positive [40,41]. A survival advantage of CLND over observation remains, however, unclear due to the lack of randomized data [40,41,46,48].

### 5.2. Radiation Therapy

MCC is a highly radiosensitive tumor. aRT of the primary tumor bed is recommended after WLE [40,41,42,52]. NCCN guidelines suggest prompt initiation of radiotherapy of the primary tumor site as soon as wound healing permits, with a dosage of 50–56 Gy if resection margins are negative, 56–60 Gy if microscopically positive, 60–66 Gy if WLE was not possible, while according to EDF guidelines a dosage of 50 Gy on the tumor region and 10 more on the tumor bed is recommended. In a meta-analysis of 1254 patients whose primary tumor was treated with WLE, local 5-year recurrences were 3 times (*p* < 0.001) and regional recurrences 2 times less probable (*p* < 0.001), when aRT of the tumor bed was administered [56].

NCCN recommends irradiation of the nodal basin on a case-by-case basis [40]. aRT could be interesting in cases of potentially false-negative SLNB (i.e., head and neck tumors, absence of immunopanel), in SNLB-negative immunosuppressed patients, or if SLNB was not performed. In cases of positive SLNB, aRT of 50–60 Gy is recommended after CLND only for patients with multiple LN involvement or extracapsular extension. Irradiation in the presence of micro-metastasis does not seem to have an impact on OS, although it has been shown to improve local control, significantly decreasing the risk of locoregional recurrence [57]. Radiation therapy may be considered as an alternative to surgical treatment for the management of primary or nodal disease in patients not eligible for surgery.

Hypofractionated radiotherapy with 10 fractions of 3.5 Gy could be an interesting option for the adjuvant treatment of the tumor bed and LN basin, with the advantage of reduced morbidity and treatment durations (NCT05100095, ongoing trial).

### 5.3. Patient Follow-Up after Excision of the Primary Tumor and the Draining Lymphatic Basin

Approximately 40% of MCC recur, and 80% of recurrences occur within 2 years of the initial diagnosis [17]. The median time-to-recurrence is 8 to 9 months [58]. Distant recurrences occur in about half of the patients, whereas LN recurrences and local or in-transit recurrences occur in approximately 25% of patients each [51,58,59]. As a result, a more intense surveillance schedule is recommended for the first three years that gradually decreases afterward, reflecting the diminishing risk of recurrence after three years [40,41,42].

For the first three years, follow-up should include clinical examination with LN palpation and LN ultrasound at three- to six-month intervals [40,41,42]. The frequency of surveillance can then be reduced to 6- to 12-month intervals for the following 2 years and then annually for life [40,41,42]. Whole-body CT or PET-CT with brain MRI with contrast may be proposed for high-risk, i.e., immunosuppressed, patients annually for the first five years. The 2018 German guidelines propose cross-sectional imaging (CT or PET-CT and cranial MRI) at three-month intervals for the first two years and then semiannually for up to five years for patients with positive or indeterminate SNLB status, due to their markedly poorer prognosis [41].

The sequential serological determination of MCPyV antibodies can be included in the monitoring of MCC patients that were seropositive at baseline [40]. Rapidly rising titers can be an early indicator of recurrence [30], reducing the need for close imaging follow-up in these patients. An ongoing, prospective study assesses the role of two blood biomarkers, MCPyV T-Ag antibodies and cell-free miR-375, as a surrogate of tumor burden in a cohort of 150 European patients (NCT04705389).

## 6. Treatment of Locally Advanced and Metastatic Disease

### 6.1. In-Transit and Local Recurrences

There is no evidence-based approach for the treatment of locoregional recurrences. In a retrospective, single-center analysis, 70 patients with locoregional MCC recurrences received either surgery or chemotherapy, radiotherapy, and palliative care [59]. The 3-year locoregional recurrence-free survival was 75% and the distant recurrence-free survival was 56%. In multivariate analysis, radiotherapy was associated with a nearly 80% reduction in mortality. Nodal status at baseline and time-to-first recurrence were important predictors of distant recurrence and OS [59].

According to current guidelines, local or in-transit and LN metastases should be surgically removed, if feasible [40,41,42]. When surgery is not feasible, radiotherapy with or without chemotherapy may be used [40,41,42]. Nodal restaging seems to be beneficial in case of locoregional recurrence since as many as 25% of patients can have synchronous or subsequent LN metastasis [59].

### 6.2. Distant Metastases

#### 6.2.1. Conventional Chemotherapy

The chemotherapeutic regimens of metastatic MCC are based on the treatment of small-cell lung carcinoma due to the similar neuroendocrine properties of both neoplasms [3,60,61] and include:-Platinum agents (carboplatin or cisplatinum agents (carboplatin or cisplatin) with etoposide and topotecan.-Cyclophosphamide, often with doxorubicin/epirubicin and vincristine, or with methotrexate and 5-fluorouracil.-Paclitaxel and a variety of other agents.

MCC is a chemosensitive tumor, but responses are short-lived. MCC chemotherapy is related to limited survival, important toxicity, and frequently acquired resistance. First-line chemotherapeutic regimens present up to 70% objective response rates (ORR), with a median response duration ranging from 3 to 10 months, whereas second- or later-line regimens have an ORR of 9–45% with a mean duration of 2 months [3,60,61].

Adjuvant chemotherapy has been proposed in the past in patients at high-risk of recurrence but failed to show any survival benefit at the cost of increased toxicities [10,17]. With MCC being an immune-sensitive tumor, chemotherapy-induced immunosuppression in the adjuvant setting could even have a negative impact on long-term outcomes.

#### 6.2.2. Immune Checkpoint Inhibitors (ICI)

The expression of viral oncoproteins in MCPyV-positive cancer cells, the high mutational burden in MCPyV-negative tumors and in the tumor micro-environment, and the clinical association with immunosuppression make MCC a good candidate for immunotherapy. PD-L1 is expressed by tumor cells in MCPyV-negative tumors, while cases with MCPyV-positive tumors are characterized by tumor-infiltrating CD8+ and CD4+ T cells targeting MCPyV oncoproteins and expressing PD-L1 and PD-1 [62,63]. Restoration of anti-tumor immunity within the tumor microenvironment through the use of anti-PD-1/PD-L1 inhibitors such as Avelumab, Pembrolizumab, and Nivolumab became the standard of care in the treatment of locally advanced/metastatic MCC in the last 5 years [40]. These agents have high efficacy, especially when administered as first-line agents (50–70%), while objective responses are observed in approximately 30% of patients in second- or later-line agents (Table 1).

Avelumab, a fully human IgG1, anti- PD-L1 monoclonal antibody, received accelerated approval in 2017 in America and Europe and in 2018 in Australia and Japan, for patients with metastatic MCC regardless of treatment history, based on the results of an open-label, single-arm, multicenter, phase II trial: the JAVELIN Merkel 200 trial part A. This study included 88 patients with stage IV chemotherapy-refractory MCC who received avelumab 10 mg/kg every 2 weeks. The ORR was 33%, with 11.4% complete and 21.6% partial responses, and most importantly durable responses confirmed by a median duration of response of 40.5 months [63,64,65]. Treatment-related adverse events were reported by 76% of patients, but most were low-grade (fatigue, infusion reactions) [64,65].

In the recently published results of the JAVELIN Merkel 200 part B study, avelumab used as first-line treatment allowed objective responses in 40% and durable responses (lasting ≥6 months) in 30% of 116 patients with metastatic MCC [66]. Real-life data from 2015 to 2018, from the expanded access program (EAP) of Avelumab, confirmed an ORR of 46.7% among 240 evaluable patients with metastatic MCC and a median duration of treatment of 7.9 months [67]. Avelumab is currently evaluated as an adjuvant treatment in MCC patients with LN metastasis at a high risk of recurrence (stages IIA–IIB) (*NCT03271372*, ADAM trial).

Pembrolizumab, a humanized IgG4 anti-PD-1 monoclonal antibody, was the second ICI approved on December 2018 for the treatment of recurrent, locally advanced, and metastatic MCC, based on data of a multicenter, nonrandomized, open-label trial: KEYNOTE-017. The ORR was 56%, with 24% complete responses in 50 treatment-naive patients receiving 2 mg/kg of pembrolizumab every 3 weeks [68,69]. The 24-month OS rate was 68.7% [68]. Pembrolizumab is currently evaluated in the adjuvant setting in patients with stage I–IIIB MCC following initial surgery (NCT03712605).

Nivolumab is a fully human IgG4, anti-PD-1 monoclonal antibody. The phase I/II single-arm, open-label, CheckMate358 trial of nivolumab at a dose of 240 mg/kg every 2 weeks in 22 patients with advanced MCC showed objective responses in 71% of treatment-naïve patients and in 63% of patients with one or more previous therapies [70]. The ORR was 68% and the 3-month OS was 92%. Nivolumab administered 4 and 2 weeks before surgery allowed presurgical pathological tumor regression in 47.2% of 36 patients with high-risk MCC, with good tolerance [71]. An international, open-label, randomized, multicenter phase 2 study assessing the efficacy of adjuvant nivolumab in MCC patients with a resected primary tumor is ongoing (NCT02196961).

Ipilimumab is a human IgG1, anti-CTLA-4 monoclonal antibody that restores tumor immunity at the priming phase in the LN and has been approved since 2011 for advanced melanoma. Since the arrival of anti-PD-1/PD-L1 inhibitors, ipilimumab is no longer used as monotherapy or as a first-line agent due to its significant immune-related toxicity. In a small retrospective study, 3/5 patients with avelumab-refractory metastatic MCC responded to the combination of ipilimumab and nivolumab [72]. In an ongoing phase 2 trial, patients were randomized to receive either nivolumab and ipilimumab (Arm A) or the combination with stereotactic body radiation therapy (Arm B), as a second-line treatment. Preliminary encouraging data on 16 of 50 included patients are available, showing an ORR of 80% for Arm A versus 17% for Arm B, with an acceptable safety profile [73]. A phase 2 trial of ipilimumab as an adjuvant therapy failed to demonstrate a significant improvement in recurrence-free survival after a median follow-up of almost 2 years, with substantial toxicity in the treatment arm, leading to its premature interruption [74].

Appendix A depicts all currently available clinical trials for ICI in combination with targeted or immune-based therapies for advanced unresectable or metastatic MCC, and Appendix A shows all available clinical trials with ICI.

In patients with primary or acquired resistance to anti-PD-1/PD-L1 inhibitors, different strategies are being evaluated:Association with radiotherapy: Radiotherapy could induce an immunogenic cell death, potentiating the effect of ICI. In two patients with progressive metastatic MCC on anti-PD-1 inhibitors, single-fraction palliative radiotherapy induced durable in-field and abscopal responses [75]. Two ongoing, phase II trials evaluate the association of either pembrolizumab or nivolumab and ipilimumab, respectively, with stereotactic body radiation therapy in metastatic MCC patients (NCT03304639, NCT03071406) [73].Switching between different anti-PD-1/PD-L1 inhibitors or adding ipilimumab could overcome resistance to ICI [72,76]. In a case series of 13 ICI refractory patients, sequentially administered salvage therapy with anti-CTLA4 alone or in combination with an anti-PD1/PD-L1 inhibitor allowed objective responses in 31% of cases [76].Combination of ICI with targeted or other immune-based therapies (see Appendix A).

To date, there are no available data concerning the appropriate duration of ICI treatment after the achievement of disease control and the optimal therapeutic sequences. Patients and physicians avoid stopping treatment due to fear of relapse, whereas over-treatment may increase toxicity and be inefficient. Ongoing strategic clinical trials seek to evaluate the length and optimal schedule of ICI treatments in responders with solid tumors, including MCC (NCT04157985, NCT05078047).

Head-to-head studies comparing the efficacy and safety of different immunotherapies for the management of locally advanced or metastatic MCC are lacking. To date, avelumab and pembrolizumab remain the only ICIs with FDA and EMA approval for this indication. A recent systematic review and meta-analysis summarized the efficacy and safety of ICI in patients with MCC [77]. Six clinical trials of a total of 201 patients were included [77]. First-line treatment with nivolumab and pembrolizumab showed a higher ORR (68% and 57%, respectively) compared to the avelumab study (ORR 41%) involving patients with prior systemic therapies [77]. These results could reflect the tendency towards better ORR and survival outcomes when ICIs are used in the first-line setting [77]. Furthermore, better responses in the pembrolizumab study could be due to the inclusion of some stage IIIB patients, while in the avelumab study all patients were stage IV. In terms of safety, patients treated with nivolumab had the fewest adverse events but had the shortest FU, once again limiting the extrapolation of these results [77].

There is an unmet need for predictive biomarkers of response to ICI in MCC. Different potential predictors have been assessed in MCC, such as clinical features, serum markers, PD-L1 expression, TIL, and genetic alterations [78]. Among clinical factors, a better performance status, the absence of immunosuppression, the early line of therapy, and a body mass index > 30 kg/m^2^ seem associated with a favorable response [78]. Kacew et al. recently suggested that a good clinical response correlated with less advanced stages of disease at baseline and shorter disease-free intervals between initial treatment and recurrence [79]. PD-L1+ expression (≥1%, evaluated with immunohistochemistry) and MCPyV tumoral status do not seem to have prognostic value, confirming the complex mechanisms of action of ICI [63,65,68,77,79]. On the other hand, PD-1 and PD-L1 density, PD-L1/PD-1 proximity, and the presence of TIL with a low T-cell receptor (TCR) clonality and a high TCR diversity seem to be positive predictors [63,80]. A recent study in multiple metastatic tumors has identified an important mutational burden and low copy-number alterations as independent response predictors [81]. Unfortunately, patients with MCC were not included in this study [81].

#### 6.2.3. Promising Novel Therapies

A non-negligible percentage of patients present a primary or acquired resistance or a contraindication to ICI, i.e., organ transplant, HIV, or severe autoimmune disease. For these patients, treatment options likely to result in durable responses are limited and inclusion in clinical trials is warranted [40]. All currently ongoing clinical studies for the treatment of advanced unresectable or metastatic MCC with targeted therapies and immune-based strategies are resumed in Appendix A.

##### Targeted Therapies

Targeted therapies seem to provide limited but durable results with a good tolerance profile in a subset of patients and could be considered in the therapeutic armamentarium of metastatic MCC in selected, ICI-refractory cases. Furthermore, the combination of targeted therapies with immune-based treatments could help through a synergistic action to optimize outcomes.

##### Vascular Endothelial Growth Factor (VEGF) Receptor Inhibitors

Pazopanib and cabozantinib are multitarget tyrosine kinase inhibitors that block VEGFR and c-kit signaling pathways, necessary for tumor angiogenesis, growth, and survival. Pazopanib also inhibits the platelet-derived growth factor (PDGF) receptor. These molecules can also modulate the tumor microenvironment by reducing the number and function of Treg cells [82]. An immunohistochemistry study of 32 MCC tumors found the expression of VEGF-A in 91% of samples, VEGFR-2 in 88%, VEGF-C in 75%, and PDGF-alpha in 72%, but only a mild expression of c-kit (7%) and PDGF-beta (13%) and no expression of epidermal growth factor receptor or Her-2/Neu [83]. These results along with encouraging data from case reports/series supported the use of pazopanib and cabozantinib in clinical trials for metastatic MCC [84]. In a phase 2, multicenter, single-arm study, 9 out of 16 patients with metastatic MCC treated with pazopanib of 600 or 800 mg daily had a clinical benefit, with a median PFS of 3.2 months and a median OS of 6.4 months. The study was stopped prematurely due to low accrual [85]. Similarly, a single-arm, phase 2 study with cabozantinib in patients with recurrent/metastatic MCC was stopped prematurely due to poor tolerability and a lack of efficacy [86].

##### KIT Inhibitors

Most MCC tumors express KIT but not activating mutations in the KIT gene’s hot-spot regions (exons 9, 11, 13, and 17) [83,87]. Co-expression of KIT and its ligand, stem cell factor, in 75% of MCC tumors suggested an autocrine activation of KIT as a possible early event in MCC transformation and differentiation [88]. Blockade of KIT and the downstream signaling cascade resulted in inhibition growth in an MCC-1 cell line in vitro [89]. In the clinical setting, however, imatinib mesylate, a c-kit inhibitor, failed to demonstrate efficacy, with only 1 among 25 patients with metastatic/unresectable MCC experiencing a partial response and a median PFS of 1 month [90].

##### Somatostatin Analogs

MCC is a poorly differentiated, high-grade neuroendocrine tumor with some degree of expression of somatostatin receptors in 85% of patients that could subsequently benefit from somatostatin analog treatments [91]. In a retrospective series of 7 patients with metastatic MCC treated with octreotide and lanreotide, 3 experienced disease control with a median PFS of 237 days [91]. There was no correlation between somatostatin receptor expression and clinical response [87]. A prospective, phase 2 study, analyzed the efficacy of lanreotide 120 mg subcutaneously every 28 days in 35 patients with metastatic/locally advanced MCC [92]. The study failed to reach predefined criteria of efficacy, namely a success rate of more than 20% at 3 months, and data analysis ended prematurely (NCT02351128) [92]. Lutetium-177-DOTATATE, a peptide receptor radionuclide therapy, can deliver radiation to tumor cells expressing the somatostatin receptor and is currently tested in metastatic MCC alone or in combination with avelumab (NCT04276597, NCT04261855).

##### PI3K/mTOR Inhibitors

Studies of genotypic profiling of MCC tumors identified a relatively low prevalence of activating somatic mutations in the PIK3CA gene (7 out of 60 samples) [89]. MCPyV-positivity and PIK3CA mutations seemed to be mutually exclusive [93]. In a MCC cell line, the targeted inhibition of PI3KCA-activating mutations resulted in increased apoptosis [93]. A patient with PIK3CA-mutated metastatic MCC achieved a complete remission at 3 months of idelalisib treatment, a selective PI3Kδ inhibitor [94]. PI3K pathway activation could drive tumorigenesis in a subset of MCC tumors and could be explored in future clinical trials.

##### Domatinostat

Domatinostat, an enzyme histone deacetylase inhibitor, is evaluated in combination with avelumab in patients with advanced MCC progressing under anti-PD-1/PD-L1 therapy (NCT04393753). One of the immune-escape mechanisms of MCC is the downregulation of MHC class I surface expression [95,96]. Histone deacetylase inhibitors have been shown to reverse low MHC class I expression epigenetically and restore tumor cell recognition and elimination by cytotoxic T cells [95].

##### Immune-Based Strategies

Intralesional treatments:

Current translational and clinical evidence support the emerging role of intralesional therapies as direct modifiers of the tumor microenvironment in MCC [97,98,99]. Intra-tumoral delivery of IFNb in MCC has been shown to upregulate MHC class I expression, restoring antigen presentation [96]. Intralesional interleukin-12 or toll-like receptor (TLR) agonists such as TLR-4 agonist and G100 have been used with promising results [97,98,99], and TLR-9 (CMP-001) and TLR-7/8 (NKTR-262) agonists are currently evaluated alone or synergistically with anti-PD-1 for metastatic MCC (NCT04916002, NCT03435640).

Talimogene laherparepvec (T-VEC) is commercially available for advanced melanoma. T-VEC is an oncolytic herpes virus, administered intratumorally, that replicates selectively within tumors with reported efficacy in a case series of patients with in-transit or locoregional MCC metastasis [100]. Phase 1/2 studies are currently evaluating T-VEC for the treatment of MCC alone (NCT03458117), with hypofractionated radiotherapy (NCT02819843), or with nivolumab (NCT02978625). RP1, another genetically modified herpes simplex 1 virus, is currently tested in organ transplant recipients with advanced MCC (NCT04349436). Localized oncolytic virotherapy can have an abscopal effect on distant non-injected lesions and could help overcome systemic tumor resistance to ICI [96].

Therapeutic vaccines:

Immunogenic epitopes clustered along the sequences of LT, ST, and VP1 have been expanded in MCPyV-positive MCCs [29,101]. Therapeutic vaccines targeting MCPyV-LT (ITI-3000) and VP1 antigens have mounted antigen-specific CD4+ and CD8+ T cell responses and prevented tumor recurrence in mice models [102,103]. A phase 1 clinical trial with ITI-3000 in the adjuvant setting is planned, with the main objective being the reduction of recurrences without significant toxicities. IFx-Hu2.0 is a plasmid DNA vaccine encoding for an immunogenic streptococcal membrane protein, Emm55, expressed by tumor cells after intra-tumoral injection. A phase I trial with IFx-Hu2.0 monotherapy (NCT04160065) and an EAP are currently available for patients with advanced/refractory MCC (NCT04853602). PD-1 blockade with simultaneous, but not sequential, T cell antigen priming with cancer vaccines could help reverse anti-PD-1 resistance [104].

Adoptive cell therapy:

Adoptive cell therapy with TIL or gene-modified T cells expressing novel TCR or chimeric antigen receptors (CAR) has proven particularly effective, especially against virus-driven malignancies. A phase 1/2 trial compared avelumab and interleukin-2 with (*n* = 1, group 1) and without (*n* = 7, group 2) infusions of HLA-restricted, MCPyV-T antigen-specific, polyclonal, autologous CD8+ T cells after tumor conditioning with radiotherapy or intra-tumoral IFNb (NCT02584829). After 1 year, 1 patient in group 1 and 4 patients in group 2 presented objective responses [105]. A promising future perspective for allogeneic adoptive T-cell transfer regardless of the HLA genotype arises from the recent expansion of MCPyV-T antigen-specific T cells from healthy donors [106]. A phase 1/2 clinical trial evaluates the efficacy of CAR T cells targeting glypican-3 (GPC3+), an oncofetal tumor antigen, and co-expressing exogenous GOT2, a mitochondrial enzyme in glutamine metabolism that contributes to cellular redox balance in metastatic, GPC3+ MCC (NCT05120271). Adoptive T cell therapies can be limited by the activation of immune-escape strategies by the tumor, namely the downregulation of the MHC class I molecules, significant toxicity (cytokine release syndrome in CAR T cell therapy), and manufacturing difficulties. Adoptive cell transfer of activated autologous or allogeneic NK cells can overcome these limitations and is under investigation in clinical trials for metastatic MCC (NCT03228667, NCCT03841110, NCT05069935, NCT02465957).

Next-generation immune checkpoint inhibitors:

Apart from PD-1 and CTLA-4, a variety of stimulatory and inhibitory coreceptors can regulate T cell activation and serve as potential drug targets. Next-generation co-inhibitory receptors blockade with anti-lymphocyte activation gene-3 (LAG-3), anti-T cell immunoglobulin and mucin-domain containing-3 (TIM-3), and anti-T cell immunoglobulin and ITIM domain (TIGIT) antibodies are evaluated either as single agents or in association with anti-PD-1/PD-L1 inhibitors in several malignancies, including MCC (NCT03538028, NCT03652077, NCT03628677). Accordingly, immunotherapies that activate costimulatory receptors of the tumor necrosis factor receptor superfamily on T cells such as OX-40 and glucocorticoid-induced tumor necrosis factor receptor (GITR) can enhance T cell differentiation and cytolytic functions and could have clinical implications in the management of metastatic MCC (NCT03071757, NCT03126110) [107].

## 7. Conclusions

The era of immunotherapy and ICI has revolutionized the treatment paradigm of locally advanced/metastatic MCC. However, primary and acquired resistance or contraindication to ICI are relevant issues for a significant subset of patients, highlighting the need for innovative therapeutic approaches. Chemotherapy is currently considered to have a palliative role, and responses, when present, are short-lived. The great diversity in the oncogenesis of MCC can explain the failure to identify tumor-specific driver mutations in most patients and the limited efficacy of targeted therapies as monotherapy. Current research priorities aim to improve the responsiveness to immunotherapy by activating immune-negligent tumors and focus on the tumor microenvironment and on combination strategies between “classic” ICI and next-generation ICI, radiotherapy, targeted therapies, intra-tumoral vaccines, and adoptive cell therapy. Other key questions that will hopefully be addressed in the near future are the appropriate treatment duration and dosage of anti-PD-/PD-L1 inhibitors among responders and the identification of predictive biomarkers of sensitivity to ICI.

## Figures and Tables

**Figure 1 cancers-15-00103-f001:**
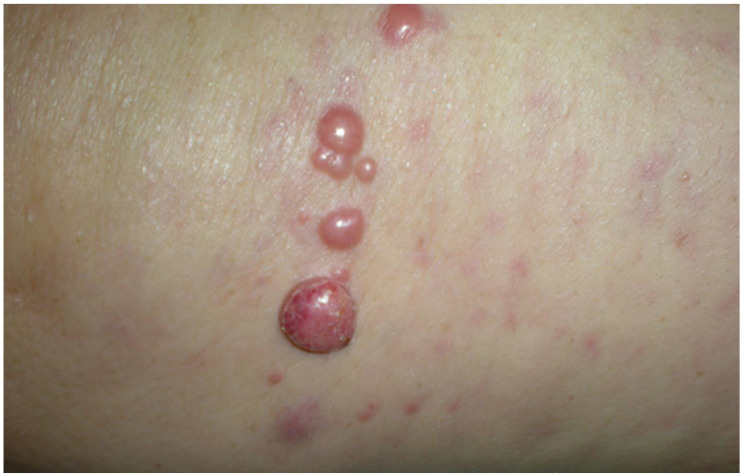
Locoregional metastases in a patient with MCC.

**Figure 2 cancers-15-00103-f002:**
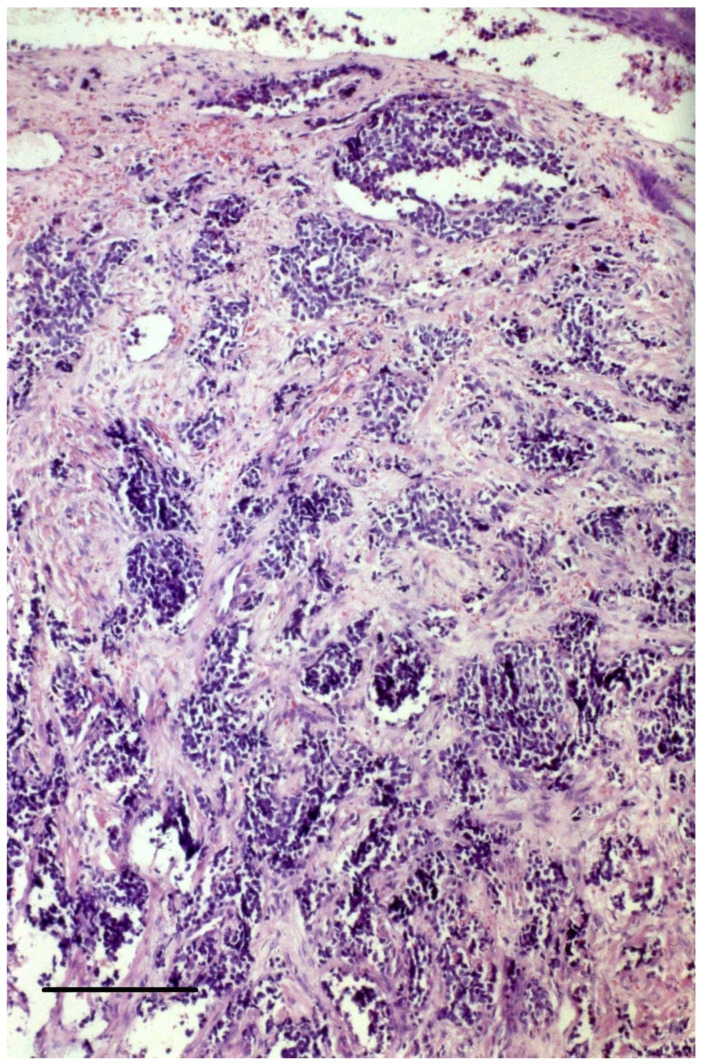
Hematoxylin-eosin stain of a primary MCC shows nests and loosely aggregated small round blue cells in the dermis. Magnification = 10 × 25. Bar = 100 μm.

**Figure 3 cancers-15-00103-f003:**
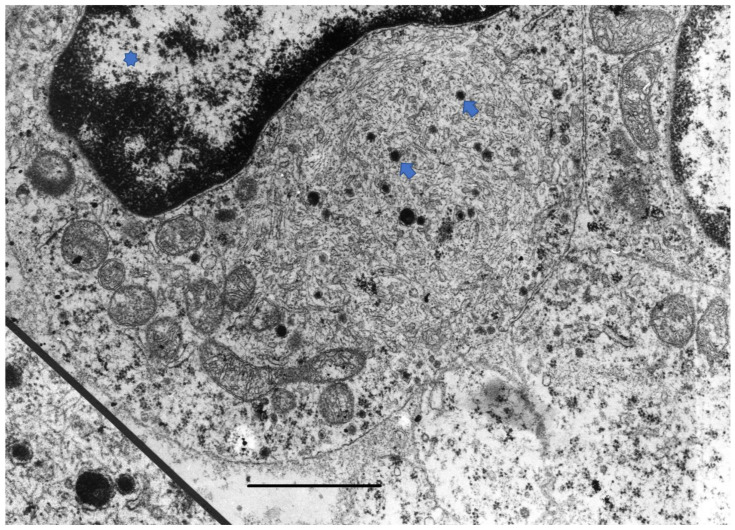
Electron microscopy of a MCC tumor cell * demonstrates electron-dense neuroendocrine granules in the cytoplasm. Asterisk: cell nucleus, Arrows: neuroendocrine granules. Magnification = ×28,980 and ×56,350. Bar = 500 nm.

**Table 1 cancers-15-00103-t001:** Efficacy results from clinical trials evaluating anti-PD-1/PD-L1 inhibitors for the treatment of locally advanced and metastatic MCC. N: number of included patients, EAP: expanded access program, ORR: overall response rate, CR: complete response, PFS: progression-free survival, OS: overall survival, FU: Follow-up, TRAE: treatment-related adverse events, m: months, NA: not available.

Drug	Avelumab			Pembrolizumab	Nivolumab
Study	JAVELIN part A	JAVELIN part B	EAP	KEYNOTE-017	CHECKMATE358
Phase	2	2		2	1/2
Regimen	10 mg/kg every 2 weeks			2 mg/kg every 3 weeks	240 mg/kg every 2 weeks
N	88	116	240	50	22
Study Duration (m)	20	36	39	12	8
Line	≥2	1	≥1	1	≤3
ORR	33%	39.7%	46.7%	56%	68%
CR	11.4%	16.4%	22.9%	24%	14%
Median PFS (m)	2.7	4.1	NA	16.8	3-month PFS 82%
Median OS (m)	12.6	20.3	NA	Not reached at 24 months	3-month OS 92%
Median FU (m)	40.8	21.2	NA	14.9	6.5
TRAEs	70%	81%	NA	98%	68%
Grade 3–4	11.4%	18.1%	NA	30%	20%
References	64, 65	66	67	68, 69	70

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
