# Peer review of "Merkel Cell Carcinoma—Update on Diagnosis, Management and Future Perspectives"

_cancers, 2022, doi:10.3390/cancers15010103_

Round 1
Reviewer 1 Report
In this review, Zaggana et al. reviewed Merkel cell carcinoma (MCC) pathogenesis, diagnosis, and management. In particular, the authors summarized the use of immunotherapy, targeted therapies, and the future therapeutic perspectives in the neoadjuvant setting and for locally advanced and metastatic MCC. The manuscript is well-organized, and I enjoyed reading it. I have a few comments to be addressed before publishing.
1. In section 3.1, is there any size range for the MCC?
2. In Figure 3, it will be helpful to label the cellular structures and point to the neuroendocrine granules with arrows.
3. In Table 1, how many treatments did the patients receive?
4. The authors may compare the effects of Avelumab, Pembrolizumab, Nivolumab, and Ipilimumab.
Minor points:
1. Figures 2 and 3 lack of scale bar.
2. In section 4.3, it is labeled as 4.3.8.
3. In 5.1, please define WLE.
4. The supporting Tables lack headers.
Reviewer 2 Report
The Authors provide a thorough look at MCC.
The Review is very up-to-date and superbly written.
Images are high quality (electron microscopy particularly is gorgeous, and very impactful too).
1. The research is a Review on Merkel Cell Carcinoma, it does not cover a specific clinical question, rather provides a general overview of the topic. 2. It is relevant in the field of Dermato-oncology as it provides readers with an up to date overview on the management of a rare but lethal skin cancer. 3. It is a Review article, so by definition, rather than adding something to the literature, it summarizes current evidences neatly and effectively. 4. The methodology is sound. 5. The conclusions are indeed consistent with the manuscript as a whole.My only suggestion is to further underscore the complexity of finding reliable response predictors (refer to and cite: Zelin E, Maronese CA, Dri A, et al. Identifying Candidates for Immunotherapy among Patients with Non-Melanoma Skin Cancer: A Review of the Potential Predictors of Response. J Clin Med. 2022;11(12):3364. doi:10.3390/jcm11123364).
No further comments on my end.
Great job.
